# Child marriage and its association with morbidity and mortality of under-5 years old children in Bangladesh

Md. Moyazzem Hossain[1]*, Faruq Abdulla[2], Rajon Banik[3], Sabina Yeasmin[4], Azizur Rahman[5]

1 Department of Statistics, Jahangirnagar University, Savar, Dhaka, Bangladesh, 2 Department of Applied Health and Nutrition, RTM Al-Kabir Technical University, Sylhet, Bangladesh, 3 Department of Public Health and Informatics, Jahangirnagar University, Savar, Dhaka, Bangladesh, 4 Statistics Division, Bangladesh Bank, Dhaka, Bangladesh, 5 School of Computing, Mathematics and Engineering, Charles Sturt University, Wagga Wagga, NSW, Australia

* hossainmm@juniv.edu

## Abstract

### Introduction

Child marriage is a significant social and health concern in many low- and middle-income countries (LMICs). This harmful practice violates children's rights and continues to be widespread across developing nations like Bangladesh. This study investigated the mortality trend among Bangladeshi children and the impact of child marriage on under-5 children morbidity and mortality in Bangladesh.

### Methods and materials

A sample of 8,321 children under-5 years old was analyzed using bivariate and multivariate statistical techniques collected from the recent 2017–18 BDHS data. Chi-square test and logistic regression (unadjusted and adjusted) were used to determine the influence of covariates on the target variable.

### Results

Results revealed that child mortality was significantly higher among children whose mothers married at an early age than their counterparts. Although the general trend in the prevalence of different childhood mortality in Bangladesh was declining gradually from 1993 to 2018, it was still high in 2018. Also, marriage after 18 years lessens likelihood of diarrhea (adjusted OR = 0.93; 95% CI: 0.76–1.16) and cough (adjusted OR = 0.91; 95% CI: 0.78–1.17) among children. Furthermore, findings reveal that likelihood of different child mortality is higher among early married women.

### Conclusion

Immediate intervention through rigorous enforcement of policies and different programs to raise the age at marriage and by lessening socioeconomic disparities can combat the

**Data Availability Statement:** After registration, the data set is accessible via the following access link http://dhsprogram.com/data/available-datasets.cfm.

**Funding:** The authors received no specific funding for this work.

**Competing interests:** The authors have declared that no competing interests exist.

prevalence of high morbidity and mortality of under-5 years old children. Findings from this study will be helpful to accelerate strategies for achieving the Sustainable Development Goals (SDGs) related to child and maternal health by 2030.

## Introduction

Child marriage is a major problem in low- and middle-income countries, as it violates human rights and disproportionately affects girls [1–3]. Unfortunately, this harmful practice remains widespread [4]. The UNICEF stated that child marriage dropped globally in the last decade; however, about 12 million girls per year even now get married before the age of 18 years [5]. The highest number of child marriages was reported in Sub-Saharan Africa (38%), followed by South Asia (30%) and Latin America (25%) [5]. For example, in Bangladesh, 59% and 22% of women were getting married before 18 and 15 years, respectively. Also, 36% of women were married before 18 years in the Dominican Republic and Brazil. In Latin America, the records are also alarming [6]. Moreover, among the Asian Muslim country, 39.5% of women had been married before the age of 18 in Pakistan [7], in 2018, approximately 11% child marriage is observed in Indonesia [8], and at least one in three young girls were married before their age of 18 years in Afghanistan [9].

Child marriage is a frequent practice in Bangladesh. Despite the advancements in recent decades, Bangladesh still has the highest percentage of child marriage in Asia and the fourth-highest rate worldwide [10]. Currently, six in every ten Bangladeshi girls are married before reaching their 18th birthday [11]. Moreover, girls who get married are often under pressure to give birth to a child early [11]. As a result, the rates of early childbearing remain high, with about 31% of Bangladeshi girls becoming mothers before they are 18 years [11]. Thus, child marriage affects them both physically and mentally and their newborns especially under-5 years children with a higher incidence of morbidity and mortality [12, 13]. Although considerable progress has been achieved in reducing under-5 child mortality, its burden remains at unacceptably excessive levels worldwide [14, 15]. About fifteen thousand children under-5 years of age died each day globally in 2017. However, the mortality of under-5 children has lessened from 12.6 million to 5.4 million from 1990 to 2017 [15].

Millennium Development Goals (MDGs) era had finished by end of 2015, during which the under-5 mortality rate decreased by 53% globally [16]. The UN Millennium Development Goal 4 (MDG-4) intends to reduce the fatality of younger children than five years by two-thirds between 1990 and 2015, but many countries, particularly in South Asia and sub-Saharan Africa, were not on track to achieve this target [17]. Although much improvement has been made in mitigating mortality among infants and children, it persists in widespread among low-middle-income countries (LMCs). Bangladesh's government has placed a great emphasis on ending child marriage. The Sustainable Development Goal (SDG)-3 is to reduce U5MR to 25 per 1000 live births in Bangladesh. Mortality among under-5 years children than five years old is the third highest in Bangladesh among South Asian countries, followed by Afghanistan. In Bangladesh, around 31 children of age under-5 died per 1000 live births in 2017, whereas it was 34 until June 2020 [18]. Socioeconomic and environmental conditions play a substantial role in the mortality and morbidity of under-5 children [19, 20]. Children living in rural areas are approximately 1.5 times more likely to die before celebrating their fifth birthday than those living in urban areas, while children from poor-income households in LMICs remain disproportionately vulnerable to early death [6, 10].

Previous studies pointed out that child marriage is a severe problem that affects adolescent girls' economic, social, and health status [12, 21–24]. A study pointed out that among middle-aged Indian women, early marriage was linked to worse self-rated health, a higher risk of functional limits, and chronic disorders [25], and it is more frequent in rural areas [26]. A significant percentage of women married before they became 18 years old, exposing them to early pregnancy and putting them in the 'thin' group (BMI below 18.5) [27]. Moreover, women married before 18 years have less chance to continue studies after marriage [28]. The age of marriage of a mother is associated with several socioeconomic and demographic variables in Bangladesh [29]. Various factors make Bangladeshi girls at risk of child marriage, such as economic pressures linked to dowry payments (money paid by the bride's family to the groom) and thinking that marriage will provide girls with social and economic security. Additionally, teenage breastfeeding mothers receive less breast milk than adult breastfeeding mothers and likely have reduced colostrum content that produces antibodies against infectious agents [30, 31], which dramatically raises the likelihood of malnutrition [32] and a variety of other adverse health effects for their infant. Studies have already shown that low socioeconomic status, unhygienic environmental conditions, and many other factors raise the risk of different infectious diseases among newborns [20, 33, 34]. Marriage in teenagers also contributes to reduced body mass index and iron deficiency [27, 35].

Furthermore, child marriage leads to numerous adverse health consequences among early married women, such as miscarriage, stillbirth, high fertility, and maternal morbidity [13, 36]. However, evidence from previous studies found a weak relationship between child marriage and under-5 mortality as well as morbidity of children [13, 23], and a study pointed out that delivery at a young age has no effect on the child's risk of under-five mortality [37]. The mortality rate of under-5 children is a significant public health concern in Bangladesh; however, evidence is deficient on the impact of child marriage on mortality and morbidity of children considering the most recent data. Therefore, there is a necessity to understand the influence of maternal age at marriage on childhood mortality and morbidity. The authors aimed to explore the trend of different types of child mortality using all BDHS data from 1993–2018 and evaluate the association between child marriage and different socioeconomic and demographic characteristics and finally determine the relationship between child marriage and mortality as well as morbidity among Bangladeshi children based on the most recent BDHS-2017-18 data.

## Methods and materials

### Data

In order to check the trend of mortality rates of children in different age groups, this study considered all available BDHS datasets (i.e. 1993–1994, 1996–1997, 2000, 2004, 2007, 2011, 2014, and 2017–18 datasets). However, for assessing the association of child marriage and determining its association with childhood morbidity and mortality, after cleaning missing data, a sample of 8,321 children under-5 years old was analyzed using bivariate and multivariate statistical techniques collected from recent 2017–18 BDHS data. The BDHS-2017-18 survey used a list of enumeration areas (EAs) of the "2011 Population and Housing Census" of the People's Republic of Bangladesh as a sampling frame. The primary sampling unit is an EA consisting of an average of 120 households. A two-stage stratified random sampling design was used to collect this survey data. About 675 EAs were chosen in the first stage, 227 and 448 EAs from urban and rural areas respectively. However, data was impossible to collect from 3 EAs (Dhaka: one urban cluster, Rajshahi: one rural cluster, and Rangpur: one rural cluster) due to a natural disaster. Therefore, a systematic sample of 30 households was chosen from every EA in the second stage. Before starting to analyze, authors use a weighted sample to ensure a country

representative sample. A detailed sampling procedure and methods of a weighted sample (mathematically adjusted) are available in the BDHS-2017/18 report [38].

## Ethical approval

This study was based on an analysis of existing public domain survey data sets that are freely available online after removing all identifier information of the respondents. This survey was approved by appropriate Ethics Committees in Bangladesh and ICF Macro at Calverton in the USA. The authors took permission from Demographic and Health Surveys (DHS) program data archivist to download the dataset for this study.

## Target variable

The primary target variable of this study is morbidity and mortality indicators of under-5 years children. The morbidity indicators included three variables: diarrhea, fever, and cough in the past two weeks. On the other hand, mortality indicators are under-five mortality (death of a child during their birth to 5 years of age), infant mortality (death of a child before his/her first birthday), neonatal mortality (death of a child before the first month of age), and post-neonatal mortality (death of a child between the first month and before the first birthday). Hence, for doing the analysis, all outcome variables were recoded as no = 0 and yes = 1.

## Independent variables

In Bangladesh, legally, the age of marriage for women is 18 years, although in the new "Child Marriage Restraint Bill 2017", marriages below 18 years are allowed [39]. However, in this study, child marriage is the primary predictor variable and is defined as a child marriage when the respondent's age at first marriage was below 18 years and an adult marriage when the respondent's age at first marriage was 18 years or more. Moreover, several demographic and socioeconomic variables are included in this study to check their association with child marriage, and in the adjusted logistic regression model, which are used as controlled variables. They are a child's sex, age, place of residence, religion, place of delivery, delivery method, division, parental education level, mother's BMI, and wealth index.

## Statistical analysis

Bivariate and multivariate analyses were carried out to measure the association between child marriage and morbidity and mortality of children below the age of 5 years. The bivariate percentage distribution examined the discrepancies in socio-demographic characteristics and morbidity and mortality indicators according to marriage groups. The Pearson chi-square test measured the significance of child, parental demographics, and child marriage compared to adult marriage. Furthermore, associations between child marriage, morbidity, and mortality of children under-5 years of age were assessed using binary logistic regression models. Relationships between variables were quantified by calculating unadjusted and adjusted odds ratios with 95% confidence intervals. In adjusted analyses, associated variables with child marriage, i.e., age of a child (eliminated from the mortality model), child's sex, and place of residence, religion, place of delivery, breastfeeding status, delivery by cesarean, father's education, mother's age, mother's educational level, mother's BMI, division and wealth index of a household were controlled. Furthermore, the Omnibus test is employed to check the significance of coefficients of a fitted logistic regression model. All data were weighted and analyzed using MS-Excel and SPSS Version 25.

### Binary logistic regression model

We utilized binary logistic regression model as defined below:

Pr(Indicator of child morbidity or mortality $= 1$)

$$= \frac{\exp(\alpha + \beta_1 \times \text{age of child} + \beta_2 \times \text{sex of child} + \ldots + \beta_k \times \text{wealth index} + \text{error})}{1 + \exp(\alpha + \beta_1 \times \text{age of child} + \beta_2 \times \text{sex of child} + \ldots + \beta_k \times \text{wealth index} + \text{error})}.$$

However, to calculate the crude odds ratio (COR), the logistic regression run by keeping only the targeted factor. Odds ratio (OR) in favor of outcome variable = 1 were computed for all covariates to suggest how many times a group of interest is more probable to belong to the target group compared to a reference group.

## Results

Selected demographic characteristics of mothers based on the categorized variable of age at first marriage were depicted in Table 1. It is seen that about 40% of mothers got married prior to the age of 18 years whose child age is less than one year. It is also observed that most of the birth are from rural areas (45.80%), Islam religion (44.51%), Rangpur division (51.00%), and from the poorest families (56.47%) with early married women compared to others. Interestingly, findings reveal that adult married mothers gave birth through health facilities and cesarean delivery. Results also reported that illiterate and less educated mothers and less educated fathers married in their early life (<18 years). They also varied for their mother's demographic characteristics. In addition, most adolescent mothers lived in rural areas (45.80% vs. 36.07%) compared to urban areas; women from lower-class families get married at an early age. Besides, a mother's BMI indicates that 64.00% of mothers are obese who do not marry at an early age. Furthermore, the prevalence of child marriage is more among illiterate women than mothers who have at least secondary education [Table 1].

The line graph illustrates different mortality rates for children in Bangladesh between 1993 and 2018. The prevalence of child mortality of all categories declined gradually throughout the study. Under-5 child mortality declined from 134 to 45 per 1,000 live births between 1993 and 2018. A similar trend is observed for infant mortality. Moreover, the infant mortality rate was 88 in 1993, then gradually decreased to 38 per 1,000 live births over the next two decades. However, in 2014 and 2018, it remains unchanged. The neonatal mortality rate also declined from 52 to 30 between 1993 and 2018. A declined scenario was also noted for post-neonatal mortality as it decreased by almost 5-folds from 1993 to 2018. Although mortality rates have been declining over the last decade, it has been slow compared to the first decade in the study period. Except for post-neonatal mortality, all are still high (45, 38, and 30 per 1,000 live births) in 2018 (**Fig 1**).

Table 2 depicts the prevalence of morbidity and mortality of under-5 years' children of married women aged between 15 to 49 years. Results revealed that among children under-5 years of age who had been affected by diarrhea in the past 2 weeks of conducting the survey, 57.58% of children who were born to early married women (<18 years old) (p-value = 0.08). About 45.13% of children of early married women had been suffered from fever in the past 2-weeks (p-value = 0.01). About 55.82% of children who were born to women who got married at an early age had experienced cough in the past 2-weeks of the survey (p-value = 0.02). Additionally, under-5 mortality (0–5 years old) was significantly prevalent among the children born to early married women (65.19%) compared to adult married women. The prevalence of infant mortality (before one year) was 62.56% for early married women and 37.44% for their counterparts. Furthermore, among children whose mothers married at an early age, 61.04% died

**Table 1. Socio-economic and demographic characteristics by child marriage (<18 years) and adult marriage (> = 18 years) in Bangladesh, BDHS-2017-18.**

| Socio-economic and demographic characteristic | Labels | Age of women at first marriage | | | | P-value |
|---|---|---|---|---|---|---|
| | | <18 Years (n = 3593) | | > = 18 years (n = 4728) | | |
| | | n | % | n | % | |
| Age of child (in months) | 0–11 | 701 | 39.47 | 1075 | 60.53 | <0.001 |
| | 12–23 | 723 | 43.04 | 957 | 56.96 | |
| | 24–35 | 702 | 42.47 | 951 | 57.53 | |
| | 36–47 | 693 | 43.48 | 901 | 56.52 | |
| | 48–59 | 774 | 47.84 | 844 | 52.16 | |
| Sex of child | Male | 1869 | 43.06 | 2471 | 56.94 | 0.082 |
| | Female | 1724 | 43.31 | 2257 | 56.69 | |
| Place of residence | Urban | 809 | 36.07 | 1434 | 63.93 | <0.001 |
| | Rural | 2784 | 45.80 | 3294 | 54.20 | |
| Religion | Muslim | 3406 | 44.51 | 4246 | 55.49 | <0.001 |
| | Non-Muslim | 187 | 28.0 | 482 | 72.0 | |
| Place of delivery | With Health Facility | 874 | 34.25 | 1678 | 65.75 | <0.001 |
| | Respondent's Home | 1252 | 48.89 | 1309 | 51.11 | |
| Currently breastfeeding | No | 1483 | 43.58 | 1920 | 56.42 | <0.001 |
| | Yes | 2110 | 42.89 | 2809 | 57.11 | |
| Delivery by C-section | No | 1596 | 46.94 | 1804 | 53.06 | <0.001 |
| | Yes | 527 | 30.93 | 1177 | 69.07 | |
| Father's education | No education | 656 | 53.86 | 562 | 46.14 | <0.001 |
| | Primary | 1432 | 50.94 | 1379 | 49.06 | |
| | Secondary and above | 1448 | 34.72 | 2722 | 65.28 | |
| Mother's Education | No education | 317 | 53.37 | 277 | 46.63 | <0.001 |
| | Primary | 1261 | 53.36 | 1102 | 46.64 | |
| | Secondary and above | 2015 | 37.57 | 3348 | 62.43 | |
| Mother's BMI | Thin (<18.5) | 559 | 49.38 | 573 | 50.62 | <0.001 |
| | Normal (18.5–24.9) | 2210 | 45.30 | 2669 | 54.70 | |
| | Obese (25 and more) | 779 | 36.00 | 1386 | 64.00 | |
| Division | Barisal | 210 | 45.55 | 251 | 54.45 | <0.001 |
| | Chittagong | 739 | 42.30 | 1008 | 57.00 | |
| | Dhaka | 791 | 37.47 | 1320 | 62.53 | |
| | Khulna | 372 | 48.44 | 396 | 51.56 | |
| | Mymensingh | 337 | 47.80 | 368 | 52.20 | |
| | Rajshahi | 491 | 50.57 | 480 | 49.43 | |
| | Rangpur | 448 | 51.00 | 480 | 49.00 | |
| | Sylhet | 205 | 30.19 | 474 | 69.81 | |
| Wealth index | Poorest | 1008 | 56.47 | 777 | 43.53 | <0.001 |
| | Poorer | 836 | 49.41 | 856 | 50.59 | |
| | Middle | 683 | 43.53 | 886 | 56.47 | |
| | Richer | 663 | 40.06 | 992 | 59.94 | |
| | Richest | 404 | 24.91 | 1218 | 75.09 | |

before their first month of life compared to 38.96% died who were born to adult married women. However, in the case of post-neonatal mortality, we observe adverse results, i.e., prevalence is lower among children born of early married women (32.06%) than adult married women [Table 2].

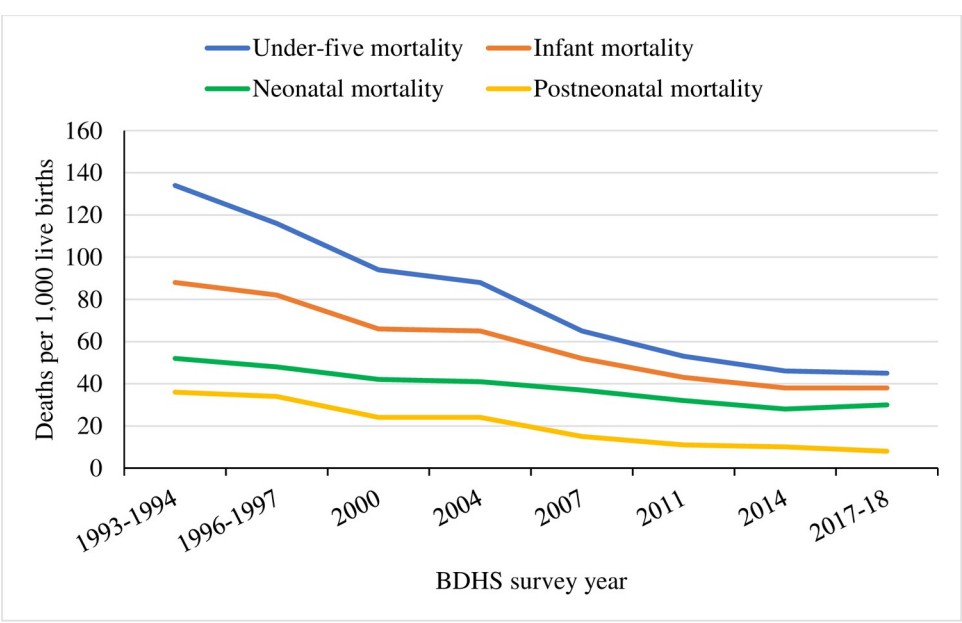

**Fig 1. Trend of mortality rates of different groups of children from 1993 to 2018, Bangladesh.**

Age of a child has been identified as a significant determinant of diarrhea. Therefore, to see a clear picture of the prevalence of diarrhea among under-5 children, Fig 2 illustrates diarrhea prevalence by age of a child and whether their mothers married as early age (<18 years old) or not. It is seen that the highest number of children suffer from diarrhea between the age of 12 to 23 months. However, after two years of age, the prevalence of diarrhea is declining for children whose mothers were getting married at an early age or not (Fig 2). Moreover, the

**Table 2. Prevalence of morbidity and mortality of under-5 years children by child marriage (<18 years) and adult marriage (> = 18 years) in Bangladesh, BDHS-2017/18.**

| Morbidity and Mortality Indicators | Labels | Age of women at first marriage | | | | P-value |
|---|---|---|---|---|---|---|
| | | <18 Years | | > = 18 years | | |
| | | n | % | n | % | |
| **Morbidity indicators** | | | | | | |
| Had diarrhea in last 2-weeks | No | 4501 | 56.77 | 3426 | 43.23 | 0.08 |
| | Yes | 227 | 57.58 | 167 | 42.42 | |
| Had fever in last 2-weeks | No | 2344 | 42.23 | 3214 | 57.77 | 0.01 |
| | Yes | 1249 | 45.13 | 1514 | 54.87 | |
| Had cough in last 2-weeks | No | 3059 | 57.36 | 2268 | 42.64 | 0.02 |
| | Yes | 1669 | 55.82 | 1325 | 44.18 | |
| **Mortality indicators** | | | | | | |
| Under-5 mortality | No | 2142 | 62.94 | 1261 | 37.06 | 0.03 |
| | Yes | 367 | 65.19 | 196 | 34.81 | |
| Infant mortality | No | 572 | 65.82 | 297 | 34.18 | 0.08 |
| | Yes | 1938 | 62.56 | 1160 | 37.44 | |
| Neonatal mortality | No | 1034 | 66.80 | 514 | 33.20 | <0.001 |
| | Yes | 1476 | 61.04 | 942 | 38.96 | |
| Post-neonatal mortality | No | 1239 | 37.71 | 2047 | 62.29 | 0.01 |
| | Yes | 218 | 32.06 | 462 | 67.94 | |

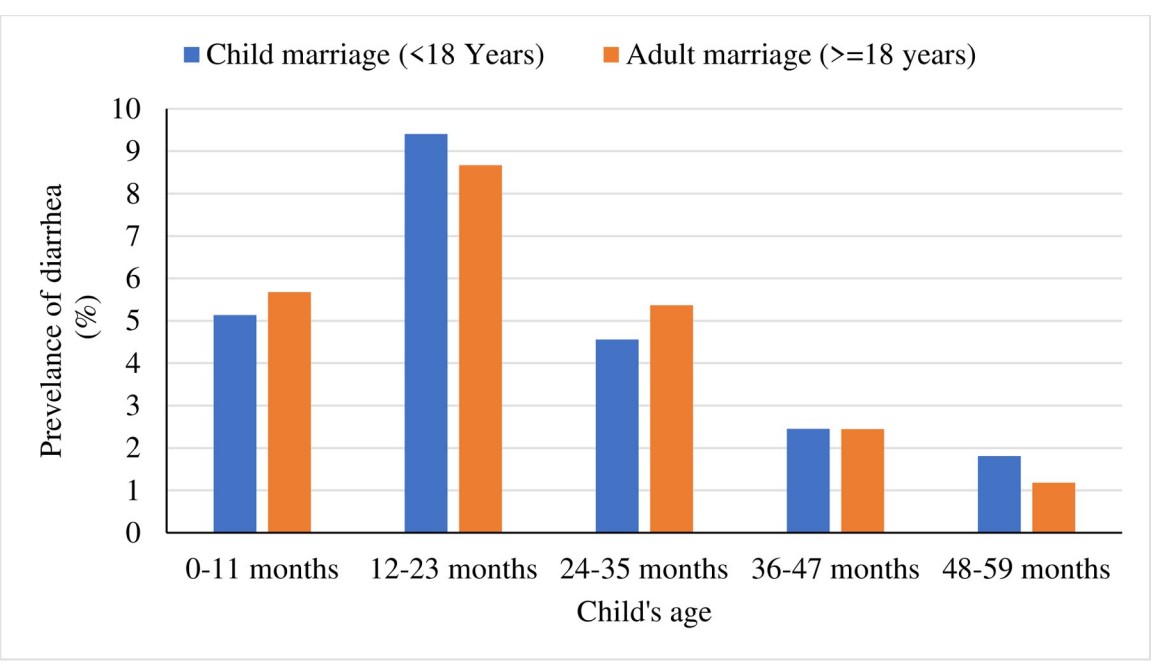

**Fig 2. Prevalence of diarrhea by child's age and status of marriage in Bangladesh, BDHS-2017/18.**

prevalence of different child morbidity and mortality indicators by selected background characteristics are presented in (S1 Table). Results depict that child morbidity and mortality indicators vary across a child's characteristics, parental characteristics, and geographical regions.

The associations between child marriage and morbidity and mortality of under-5 children are presented in Table 3. The p-value of the Omnibus test is employed to check the significance of coefficients of a fitted logistic regression model, i.e., modelling between morbidity indicators and child marriage is less than 0.001, which concludes that we may reject the null hypothesis that intercept and all coefficients are zero. Also, results of the classification table demonstrated that about 95.2%, 66.8%, and 64.1% of observations are classified accurately by fitted model for diarrhea, fever, and cough, respectively. Similarly, in the case of the fitted model for different child mortality and child marriage, results indicate that we may reject the null hypothesis that all coefficients are zero at 5% level of significance since all p-values are less than 0.001. Findings also illustrated that the classification accuracy for under-5, infant, neonatal and post-neonatal mortality are 86.2%, 78.5%, 61.5% and 83% respectively [Table 3].

Results revealed that children of adult married ($> = 18$ years old) women were less vulnerable to have diarrhea (COR: 0.97, 95% CI: 0.79–1.18) and cough (COR: 0.86, 95% CI: 0.72–1.16) except fever (COR: 1.12, 95% CI: 1.01–1.22) though it is insignificant than those children whose mother were married at an early age (<18 years old). The association also remained significant for diarrhea (AOR: 0.93, 95% CI: 0.76–1.16), cough (AOR: 0.91, 95% CI: 0.78–1.17), and fever (AOR: 1.05, 95% CI: 1.01–1.22) after controlling relevant demographic characteristics in logistic regression analysis. However, there is a 5% more chance to suffer from the fever of children of adult married women than their counterparts. Children born to women who were married at age of 18 years and later were approximately 21% less likely to under-5 child mortality (AOR: 0.79, 95% CI: 0.62–1.02) than children whose mothers were married early age. The relationship between maternal adolescent motherhood and neonatal mortality was also significant in adjusted analysis. However, in the case of post-neonatal mortality, a child from adult married women have 13% higher chance of post-neonatal mortality than a child

**Table 3. Impact of child marriage (< 18 years old) on morbidity and mortality of under-5 children in Bangladesh; BDHS-2017-18.**

| Characteristics | COR (95% CI) | AOR (95% CI) | Model validation |
|---|---|---|---|
| **Morbidity indicators** | | | |
| **Had diarrhea in last 2-weeks[a]** | | | |
| Child marriage (Ref.) | | | Omnibus Tests (p-value): <0.001 |
| Adult marriage | 0.97 (0.79–1.18)* | 0.93 (0.76–1.16)* | Classification percentage: 95.2 |
| **Had fever in last 2-weeks[a]** | | | |
| Child marriage (Ref.) | | | Omnibus Tests (p-value): <0.001 |
| Adult marriage | 1.12 (1.03–1.23) | 1.05 (1.01–1.22)* | Classification percentage: 66.8 |
| **Had cough in last 2-weeks[a]** | | | |
| Child marriage (Ref.) | | | Omnibus Tests (p-value): <0.001 |
| Adult marriage | 0.86 (0.72–1.16)* | 0.91 (0.78–1.17)* | Classification percentage: 64.1 |
| **Mortality indicators** | | | |
| **Under-5 mortality[a]** | | | |
| Child marriage (Ref.) | | | Omnibus Tests (p-value): <0.001 |
| Adult marriage | 0.92 (0.84–1.33)* | 0.79 (0.62–1.02)** | Classification percentage: 86.2 |
| **Infant mortality[a]** | | | |
| Child marriage (Ref.) | | | Omnibus Tests (p-value): <0.001 |
| Adult marriage | 0.82 (0.74–1.02)* | 0.89 (0.67–1.13)* | Classification percentage: 78.5 |
| **Neonatal mortality[a]** | | | |
| Child marriage (Ref.) | | | Omnibus Tests (p-value): <0.001 |
| Adult marriage | 0.78 (0.68–0.89)*** | 0.84 (0.72–0.97)** | Classification percentage: 61.5 |
| **Post neonatal mortality[a]** | | | |
| Child marriage (Ref.) | | | Omnibus Tests (p-value): <0.001 |
| Adult marriage | 1.12 (1.08–1.53)** | 1.13 (0.86–1.20)** | Classification percentage: 83.0 |

*Note*: COR = Crude Odds ratio; AOR = Adjusted Odds ratio; CI = confidence interval; *p < 0.05, **p < 0.01, ***p < 0.001; [a]Adjusted analyses controlled for age of the child, sex of the child, place of residence, religion, place of delivery, breastfeeding status, delivery by caesarean, father's education, maternal education, maternal BMI, division and wealth index; Ref.: Reference category.

from early married women. Findings reveal that there is 11% higher likelihood of infant mortality among children whose mothers married early than adult married women after adjusting some relevant demographic characteristics [Table 3]. Furthermore, detailed results of logistic regression for each model are summarized in a [S2 Table].

## Discussion

Child marriage is a frequent social practice in Bangladesh, and its high prevalence contributes to early pregnancy, which poses a long-term danger to younger mothers and the wellbeing of newborns [40, 41]. Initially, authors find the association of different socio-economic factors with child marriage and the effect of covariates are adjusted in logistic regression. Findings reveal that the prevalence is significantly varied by considering age and sex of a child, parental education, wealth index, residence. In Bangladesh, practicing of child marriage is significantly higher among low-income families because it is assumed that girls are specially born to run households. Therefore, it is generally thought that sending females for higher studies seems to create an economic burden. In rural areas, most households can afford to send at most two children to school, and in such cases, sons receive preference due to their role as to carry the name and family title of father and will assure parent's future [42, 43]. Nowadays, educated parents are more concerned about women's empowerment, and higher educated parents will have a higher educated child because higher educated parents are financially rich, and they

make enormous efforts to confirm their children's academic success [43–45]. Therefore, the regional difference in educational, socio-cultural, and financial status are key factors of practicing child marriage [28].

Findings presented in Fig 1 illustrate that infant, neonatal, post-neonatal, and under-5 years of child mortality rates in Bangladesh had a declining rate over the study period. This study investigated the relationship of child marriage with morbidity and mortality of under-5 years of age children in Bangladesh by exploring a trend of child mortality. This study revealed that child marriage was significantly associated with morbidity and mortality of children under-5 years of age even after adjusting socio-demographic and other covariates. The findings of this study are somewhat different from previous studies in India and Pakistan, which showed child marriage weakly correlated with under-5 years child morbidity and mortality but no correlation identified in adjusted models [13, 23, 24].

Considerable evidence suggests that child marriage has significant adverse consequences detrimental to girls, their children, and their communities. A child born to a teenage mother is twice as likely to die before the first day of her life than a mother of aged twenties [41, 46]. Evidence from earlier studies has also demonstrated that child marriage raises the incidence of infectious diseases for their newborns and thereby enhances the susceptibility of infant, child, neonatal, and post-neonatal mortality [47, 48]. Even though these children live, they are more vulnerable to low birth weight, premature birth and mortality, than children born to older mothers [49]. Prior studies in Bangladesh revealed that neonates and children with younger mothers have a greater mortality risk than older mothers [40, 41]. Plausible explanations for this could be the early age of mothers correlated with bad health-seeking behavior [50] and mother's experience with optimum infant feeding [51]. Unlikely, other research found no significant association between teenage mothers and child morbidity, such as diarrhea and fever [12, 23]. Surprisingly, this research also demonstrated that children of adult married women were significantly less vulnerable to diarrhea (AOR: 0.93) and cough (AOR: 0.91) than children born to child married women. Child marriage in Bangladesh is possibly a proxy for maternal limitations that endanger young children's wellbeing, which raises the probability that young mothers and their related social and economic shortcomings contribute to poor child health outcomes [52, 53]. Our results illustrate the importance of delaying childbirth for those married under the age of 18 years and indicate that the younger mother's socioeconomic and cultural disadvantages can contribute to poor health outcomes for children.

Over the last decades, Bangladesh has undergone a substantial decline in child mortality [54], which was helpful to gain the MDG-4 target. Nevertheless, mortality among under-5 years of old children and neonatal mortality need to be lessened for achieving the Sustainable Development Goals (SDGs), which are intended to reduce neonatal and under-five mortality rates to at least as low as 12 and 25 deaths per 1,000 live births respectively [18]. Reducing the prevalence of child marriage would be beneficial to achieve targeted SDG goals.

## Strengths and limitations

The strength of this study is the novelty of the work and considering the most recent country representative BDHS-2017/18 dataset. This analysis is cross-sectional, and causal inference is not possible; but, since child marriage occurred before the measured results, ordering of this exposure could be presumed compared to child health incidents. The examined results are based on self-reporting and are thus vulnerable to variation in memory and social desirability. Similarly, multiple socioeconomic and environmental factors in this setting may contribute to the incidence of diarrhea and related mortality that was beyond track in the regression model. This included only specific infant and maternal demographics in multivariate analyses. We

have also restricted our sample to births in the previous five years for the ever married 15–49 years old women. Lastly, results are unique to young women in Bangladesh and cannot be generalized within Bangladesh to other regional contexts. Future research would be carried out to find the effect of child marriages on child health outcomes incorporating other factors like child's BMI, immunization status, spatial variation by utilizing PSM methods and Matching techniques to get a clear comparison picture of the two groups studies.

## Conclusion

Most child-married women are from rural areas, follow the Islam religion, live in the Rangpur division, and come from the poorest families. Findings reveal that both illiterate and less-educated men and women were getting married at an early age. Although rates of several categories of child mortality have decreased dramatically over the last three decades, all are still high (45, 38, and 30 per 1,000 live births) in 2018. This study also revealed that child marriage of women marginally influences mortality and morbidity of under-5 year's children in Bangladesh. A variety of socio-demographic and cultural differences are significant predictors of this issue. However, a longitudinal study is required to recognize the impact of child marriage on the health outcomes of children. Authors suggest that the government of Bangladesh should try to provide adequate assistance and structural modifications to eradicate child marriage that will benefit girls along with their families, community, and society. Authors also recommend an upsurge in age at women's marriage by rigorous enforcement of child marriage laws. Furthermore, participation in education and empowerment of younger women may help to minimize child morbidity and mortality rates. Finally, the need for tailored intervention to sustenance children born to mothers married at an early age is crucial which can reduce morbidity and mortality among under-5 years of age children in Bangladesh, which will be helpful to accelerate strategies for achieving Sustainable Development Goals (SDGs) related to child and maternal health by 2030.

## Supporting information

**S1 Table. Prevalence of child morbidity and mortality Indicators by Socio-economic and Demographic Characteristics, BDHS-2017/18.**
(DOCX)

**S2 Table. Results of logistic regression, BDHS-2017/18.**
(DOCX)

## Author Contributions

**Conceptualization:** Md. Moyazzem Hossain, Faruq Abdulla, Sabina Yeasmin, Azizur Rahman.

**Data curation:** Md. Moyazzem Hossain.

**Formal analysis:** Md. Moyazzem Hossain, Faruq Abdulla, Sabina Yeasmin.

**Methodology:** Azizur Rahman.

**Supervision:** Md. Moyazzem Hossain, Azizur Rahman.

**Visualization:** Md. Moyazzem Hossain, Faruq Abdulla.

**Writing – original draft:** Md. Moyazzem Hossain, Faruq Abdulla, Rajon Banik, Sabina Yeasmin.

**Writing – review & editing:** Md. Moyazzem Hossain, Faruq Abdulla, Azizur Rahman.

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
