## [Editor Report · Decision Letter 0]

16 Sep 2021

PONE-D-21-24855Child marriage and its effect on morbidity and mortality of under-5 years old children in Bangladesh: evidence from a countrywide cross-sectional surveyPLOS ONE

Dear Dr. Hossain,

Thank you for submitting your manuscript to PLOS ONE. After careful consideration, we feel that it has merit but does not fully meet PLOS ONE’s publication criteria as it currently stands. Therefore, we invite you to submit a revised version of the manuscript that addresses the points raised during the review process.

ACADEMIC EDITOR:

Although the research question of this study is very important, but in its current form it doesn't suits for publication in PLOS One. The paper is less likely receive favourable opinion from the reviewers. The analytical plan and tools are very weak and doesn't support the research question. Thus, I recommend authors to revise and resubmit to PLOS One or any other journal. Moreover, dont attempt to measure impact or causation from a cross-sectional data. Try to use PSM to address the adverse affect of child marriages on child health outcomes.  Consult following papers for methods, concept and framework of analyses. 

Vikram, Kriti. "Early marriage and health among women at midlife: Evidence from India." Journal of Marriage and Family (2021).

Goli S, Rammohan A, Singh D. The Effect of Early Marriages and Early Childbearing on Women's Nutritional Status in India. Matern Child Health J. 2015 Aug;19(8):1864-80. doi: 10.1007/s10995-015-1700-7. PMID: 25656721.

Goli, S. (2016). *Eliminating child marriages in India: Progress and prospects*. Child Rights Focus-Action Aid. https://www.actionaidindia.org/publications/eliminating-child-marriage-in-india/

We look forward to receiving your revised manuscript.

Kind regards,

Srinivas Goli, Ph.D.

Academic Editor

PLOS ONE

Additional Editor Comments (if provided):

Although the research question of this study is very important, but in its current form it doesn't suits for publication in PLOS One. The paper is less likely receive favourable opinion from the reviewers. The analytical plan and tools are very weak and doesn't support the research question. Thus, I recommend authors to revise and resubmit to PLOS One or any other journal. Moreover, dont attempt to measure impact or causation from a cross-sectional data. Try to use PSM to address the adverse affect of child marriages on child health outcomes. Consult following papers for methods and framework of analyses.

Vikram, Kriti. "Early marriage and health among women at midlife: Evidence from India." Journal of Marriage and Family (2021).

Goli S, Rammohan A, Singh D. The Effect of Early Marriages and Early Childbearing on Women's Nutritional Status in India. Matern Child Health J. 2015 Aug;19(8):1864-80. doi: 10.1007/s10995-015-1700-7. PMID: 25656721.

Journal Requirements:

“No. The funders had no role in study design, data collection and analysis, decision to publish, or preparation of the manuscript.”

5. Please ensure that you refer to Figure 2 in your text as, if accepted, production will need this reference to link the reader to the figure.

6. We note that Figure 2 in your submission contain map images which may be copyrighted. All PLOS content is published under the Creative Commons Attribution License (CC BY 4.0), which means that the manuscript, images, and Supporting Information files will be freely available online, and any third party is permitted to access, download, copy, distribute, and use these materials in any way, even commercially, with proper attribution. For these reasons, we cannot publish previously copyrighted maps or satellite images created using proprietary data, such as Google software (Google Maps, Street View, and Earth). For more information, see our copyright guidelines: http://journals.plos.org/plosone/s/licenses-and-copyright.
---

## [Author Response · Author response to Decision Letter 0]

19 Sep 2021

Dear Srinivas Goli, Ph.D.

Academic Editor

PLOS ONE

We would like to sincerely thank the Academic Editor for your valuable comments. We have considered all comments and then thoroughly revised and formatted the manuscript. A detailed response to each particular comment is provided in the tables as follows.

Thank you very much for your valuable comment and feedback. We believe that it will be helpful to improve the quality of the manuscript. 

We have cited the suggested papers in our manuscript. 

Thanks for your suggesting to use PSM for finding the adverse effect of child marriages on child health outcomes. However, in light of the existing literature, it is found that there are several manuscript used the logistic regression to identify the association between child marriage and child health outcome. Therefore, we think this would be a potential topic for future research. All revised texts are in “red” color. Page 1, 4, 7-9, 14-15, 18-19.

Thanks. We ensure you that we follow the journal guidelines for formatting purpose. 

 Thank you very much for your valuable comment and suggestions that help improve the manuscript's quality. We thoroughly checked the grammatical issues and also check by Grammarly software. 

We upload the required files following the guidelines provided. All revised texts are in “red” color. Page 1-15.

Thank you very much. The authors received no specific funding for this work. We add this statement in the manuscript and cover letter. Revised texts are in red color. Page: 15.

Thanks. We move the ethical statement from the end of the manuscript to the Methods section. The revised texts are in red color. Page: 5.

Thanks. We have removed Figure 2 from our manuscript. Page: 12.

In conclusion, the revised version of the manuscript has been produced as per the review outcomes. So, we hope that you would be happy to see this greatly improved version. Once again, we would like to thank you all for your dedication, professional services and cooperation.

---

## [Decision Letter · Decision Letter 1]

25 Nov 2021

PONE-D-21-24855R1Child marriage and its association with morbidity and mortality of under-5 years old children in BangladeshPLOS ONE

Dear Dr. Hossain,

Thank you for submitting your manuscript to PLOS ONE. After careful consideration, we feel that it has merit but does not fully meet PLOS ONE’s publication criteria as it currently stands. Therefore, we invite you to submit a revised version of the manuscript that addresses the points raised during the review process.

ACADEMIC EDITOR: Considering my own reading of the paper and all four reviewers observations, I am going with a decision of Reject and Resubmit. 

We look forward to receiving your revised manuscript.

Kind regards,

Srinivas Goli, Ph.D.

Academic Editor

PLOS ONE

Additional Editor Comments:

Considering my own reading of the paper and all four reviewers observations, I am going with a decision of Reject and Resubmit.

Reviewers' comments:

Reviewer's Responses to Questions

**Comments to the Author**

1. If the authors have adequately addressed your comments raised in a previous round of review and you feel that this manuscript is now acceptable for publication, you may indicate that here to bypass the “Comments to the Author” section, enter your conflict of interest statement in the “Confidential to Editor” section, and submit your "Accept" recommendation.

Reviewer #1: (No Response)

Reviewer #2: All comments have been addressed

Reviewer #3: (No Response)

Reviewer #4: All comments have been addressed

2. Is the manuscript technically sound, and do the data support the conclusions?

Reviewer #1: Yes

Reviewer #2: Partly

Reviewer #3: Yes

Reviewer #4: Partly

3. Has the statistical analysis been performed appropriately and rigorously? 

Reviewer #1: Yes

Reviewer #2: Yes

Reviewer #3: Yes

Reviewer #4: No

4. Have the authors made all data underlying the findings in their manuscript fully available?

Reviewer #1: Yes

Reviewer #2: Yes

Reviewer #3: Yes

Reviewer #4: Yes

5. Is the manuscript presented in an intelligible fashion and written in standard English?

Reviewer #1: Yes

Reviewer #2: Yes

Reviewer #3: No

Reviewer #4: No

6. Review Comments to the Author

Reviewer #1: I have some minor suggestions. Please find them below:

1. Line 109- replace 'published BDHS datasets' with some other relevant word. 'Published' is not a correct word here.

2. In table 1 (and wherever relevant): place 'in month' in bracket after age of child. It should be- age of child (In months). Table 1 first demographic characteristic.

3. Please confirm the p-value for teh sex of the child in table 1.

4. Authors may provide CIs for crosstabs also. This would increase the robustness of the study findings.

5. Consider changing post-natal mortality to post neonatal mortality.

6. Furthermore, women who particularly lived in rural areas, are not permitted to visit a health

care facilities for their own as well as their children’s health [54,55]. In above sentence, authors cited quite old references. Either update the reference or consider omitting the sentence as the current situation may be different.

7. According to the findings of the study, illiterate and undereducated mothers and fathers married in their early age. In the sentence, consider replacing 'mothers and fathers' as when they married off, they were men and women and not mothers and fathers.

Reviewer #2: Title: Child marriage and its association with morbidity and mortality of under-5 years old

children in Bangladesh

Authors: Md. Moyazzem Hossain, Rajon Banik, Faruq Abdulla, Sabina Yeasmin and Azizur

Rahman

Thank you for the chance to review the paper. It is an interesting paper that focuses on Child marriage and its association with child morbidity and mortality pattern in Bangladesh. Overall, the paper has explored a vital area; however, for strengthening the paper, here I have outlined my concerns and suggestions.

• Writing “early married women” Instead of “child married women” sounds more appropriate. I would suggest to rewrite using the suggested terms. (Found in lines – 36, 184, 213, 214, 217, 219, 223)

• The sentence “Girls who get married are often under pressure to give birth to a child early”- in line number 59 needs reference.

• The statement in lines 81-84 consists factual information. Please provide reference supporting the information.

• The statement in lines 270-271 mention about “map”, however the map has been removed by the authors. Please modify the statement.

• The authors are suggested to incorporate “Child’s BMI” and “Child’s Immunization status” as controlled variable in the binary logistic regression model, since these variables can affect the morbidity and mortality of children under age 5. All these variables can be created using BDHS dataset.

• Reading discussion of this, I have found that most of the findings are found consistent with previous studies. Then the authors should clarify the contribution of this study to academia. More focus should be given to the unique findings first and then others.

• Moreover, I would suggest to use PSM technique rather than logistic regression. The PSM method can strengthen the analysis and can stand as more appropriate for this study.

Reviewer #3: Comments

The central theme and the issue discussed here in the paper is significant and holds relevance to the country since as stated by the authors, in Bangladesh, 59% and 22% of women get married before the age of 18 and 15 years, respectively. And with about 31% of Bangladeshi girls becoming mothers before they are 18 years, this is a serious issue and should be studied thoroughly, As such, the analysis is timely and of potential value for policymakers. Appreciate the authors for taking such a relevant issue to study. This would definitely lead to an increase in the knowledge base in this domain. However, some problems with the writing of this paper should be addressed before the manuscript is accepted for publication.

There are some more revisions required in writing. Some of the errors which might need revision are-

Line 32 Was declined- was declining

Line 33- is still high- was still high

Line 47- that affect- which in turn affect

Line 132- the authors take permission- the authors took permission

Line 189- urban areas, women who are from lower-class families get married at an early age.

Line 198 - categories was declined- categories declined

Line 199- mortality was declined- mortality declined

Line 204- was decreased- decreased

Line 215-216- moreover has been used repeatedly. The sentence should be modified.

Line 353- According to the findings of the study, illiterate and undereducated mothers and fathers married at their early age. -Rephrase the sentence.

Overall, English write-up needs to be checked carefully in the whole manuscript, sometimes some errors are overlooked by software. There should be a flow in the usage of tenses. The methodology is fine and easy to understand but it is basic, the authors could have used some more sophisticated methodology to compare and get a clear comparison picture of the two groups studies. Some Matching techniques can be used which would make the study unique and completely different from the already done work. Further, such a methodology would also help in understanding a few new dynamics of the factors contributing to child marriage and child mortality in Bangladesh.

Incorporating these few changes, this study would be a valuable resource for future studies in this field.

Reviewer #4: This article is based on cross-sectional data from a cross-sectional population survey with 8321 children under the age of five in Bangladesh. The first aim was to estimate the births to women born into child married and adult married women in Bangladesh by socioeconomic characteristics, defined as women age at first birth <18 years and >18 years. The second objective relates to the prevalence of morbidities and mortalities under 5 years’ children by child marriage and adult marriage women. The author looks at three morbidities in the last two weeks, such as diarrhea, fever, and cough, as well as four mortality indicators, including infant mortality, neonatal mortality, and post-neonatal mortality. The findings revealed that child mortality among adult married women was considerably lower. However, there are certainly major issues that must be addressed.

Introduction

Unfortunately, the language is difficult to understand. To improve the text's understandability, flow, and readability, I recommend the authors engage with a translator, writing coach, or copyeditor. Even, I can see, the editor highlighted the same concern about the writing quality, however, it hasn't significantly improved.

Results

The author has written the age at first marriage in lines 180 and 181, but the table indicates the age of the women at first birth, which may confuse the reader. Please correct the terminology. "43% of children's mothers got married before the age of 18 years”, the authors wrote again on line 181, but the table author has taken the children age groups but do not specify in the above statement. In line number 188, the authors repeated the findings from line number 182, which are related to the 45.8% of births to mothers under the age of 18 years. The author has stated "obesity is lower among women who do not marry at a young age compared to their counterparts" on line 189, however the author's unit of analysis is children, therefore please correct this. There are numerous typos in Table 1, so kindly update these names. Another observation is that the age of the child in month variables labels are not properly specified, such as 36-47 near interval, and the last label started again 47-59; please explain why. The author wrote that the COR for fever is 1.05 in line 249, however, the table shows that it is 1.12. There is an incomplete sentence in line 253 of the manuscript. The authors have written that the OR is 0.79 and that the 20% more likely is shown in line 256; please correct this sentence. The authors used early married women in line 260, but married before their 18 years’ age in line 255, which would confuse the reader. Please use the same variables names that are used in the tables.

Some general comments regarding figures and graphs

Figure 1 is not a standard graph; please make a standard line graph.

Table 2: The P-value was misdefined by the authors.

Table 3: The author applied the Omnibus test, however the methodology section did not mention it.

Discussion

"Children of women married as a child were significantly more vulnerable to diarrhea and fever than children born to married women as adults," the authors wrote in line 288, I assume that they were referring to the adjusted odds ratio. However, the result in table no 3 is completely different.

The author has written in lines 303, 304, and 305 that this research showed that infant, neonatal, post neonatal, and under -5 years of child mortality in Bangladesh increased from 1993 to 2014 over the years and the rate is diminishing steadily over time, yet these are unclear and perplexing.

The author writes in line 311 that "a substantial number of social-demographic variables have a significant effect on child married in Bangladesh," however I don't find any evidence that the authors looked into child marriage or that the author's unit of analysis were child married women.

The last paragraph of the discussion, in lines 328-329, is not the authors' results, although they do not provide any references.

I probably stick to a simple structure and using precise terms. It is necessary to improve the discussion. The authors are confused by the unit of analysis in general throughout the article. Many grammatical errors remain.

7. PLOS authors have the option to publish the peer review history of their article (what does this mean?). If published, this will include your full peer review and any attached files.

Reviewer #1: No

Reviewer #2: No

Reviewer #3: No

Reviewer #4: No

---

## [Author Response · Author response to Decision Letter 1]

8 Dec 2021

Dear Editor,

We sincerely thank the Academic Editor and four anonymous reviewers for their valuable comments. We have considered all comments and then thoroughly revised and formatted the manuscript. A detailed response to each comment is attached with this submission. 

We also hope that you will be happy to see this greatly improved version. Once again, we would like to thank you all for your dedication, professional services and cooperation.

Thanks in advance.

Hossain

---

## [Decision Letter · Decision Letter 2]

26 Dec 2021

PONE-D-21-24855R2Child marriage and its association with morbidity and mortality of under-5 years old children in BangladeshPLOS ONE

Dear Dr. Hossain,

Thank you for submitting your manuscript to PLOS ONE. After careful consideration, we feel that it has merit but does not fully meet PLOS ONE’s publication criteria as it currently stands. Therefore, we invite you to submit a revised version of the manuscript that addresses the points raised during the review process.

ACADEMIC EDITOR: Reviewers still not pleased with the revisions. Considering their opinion and my own reading, I am recommending a major revision for this manuscript. This will be a last chance for you to incorporate the reviewers suggestions. 

We look forward to receiving your revised manuscript.

Kind regards,

Srinivas Goli, Ph.D.

Academic Editor

PLOS ONE

Journal Requirements:

Additional Editor Comments (if provided):

Reviewers still not pleased with the revisions. Considering their opinion and my own reading, I am recommending a major revision for this manuscript. This will be a last chance for you to incorporate the reviewers suggestions. 

Reviewers' comments:

Reviewer's Responses to Questions

**Comments to the Author**

1. If the authors have adequately addressed your comments raised in a previous round of review and you feel that this manuscript is now acceptable for publication, you may indicate that here to bypass the “Comments to the Author” section, enter your conflict of interest statement in the “Confidential to Editor” section, and submit your "Accept" recommendation.

Reviewer #1: All comments have been addressed

Reviewer #3: All comments have been addressed

Reviewer #4: All comments have been addressed

2. Is the manuscript technically sound, and do the data support the conclusions?

Reviewer #1: Yes

Reviewer #3: Yes

Reviewer #4: Partly

3. Has the statistical analysis been performed appropriately and rigorously? 

Reviewer #1: Yes

Reviewer #3: Yes

Reviewer #4: I Don't Know

4. Have the authors made all data underlying the findings in their manuscript fully available?

Reviewer #1: Yes

Reviewer #3: Yes

Reviewer #4: Yes

5. Is the manuscript presented in an intelligible fashion and written in standard English?

Reviewer #1: Yes

Reviewer #3: No

Reviewer #4: No

6. Review Comments to the Author

Reviewer #1: The authors have carried out the revision as suggested and that has improved the manuscript to an extent. I still have some doubts and suggestions regarding the manuscript, which are as follows:

1. The image in figure 1 does not seem to be matching with the text in the manuscript. The authors can please clarify this. For ex: in text, it was written "Moreover, the infant mortality rate was 88 in 1993, then gradually decreased to 38 per 1,000 live births over the next two decades. " However, the figure is showing different numbers. Please confirm, if I am wrong in analyzing the figure. A similar mismatch is there in the text at other places too for figure 1.

2. For table 2, if authors could segregate the prevalence of diarrhea by age of the children, it would present a clear picture. Studies have noted the age of the children as an important factor in the prevalence of diarrhea. so presenting diarrhea among children of various ages by women's age at marriage.

3. Please confirm that for table 3, what is the reference category? In table 2, authors have taken two categories for married women - women who married below 18 years and women who married at 18 and above 18 years. But in table 3, authors have taken women as above 18 years old, thereby leaving women who were 18 years old. Why so discrepancy?

4. Table 1 is providing the differences by age at marriage of the women. I would also suggest to add a table (may be as supplementary) that provides within group differences. For ex: it would be interesting to see the differences between sex, religion, and other covariates. currently, the total adds up for women age grouyp. I want that to be added for covariate.

Reviewer #3: There have been various necessary changes which are praiseworthy. However, a little English revision still needs to be done. After which the paper is good to publish.

Some of the places where revision is needed-

Line 46 which in turn affecting

Line 49 marriages has happened

Line 110 there is necessary – there is a necessity

Line 156-They are a – should be –‘which are’

Line 160- The bivariate and multivariate - Bivariate and multivariate

Line 165- child marriage and morbidity and mortality- child marriage, morbidity and mortality

178- variable that assign- Variable that assigns

Paper should be thoroughly read and other errors should be rectified to standard English.

Usage of article “The” in the sentences should be checked and only put where ever necessary.

Mother’s BMI has been divided in to- Thin, Normal and Obese, does this include overweight and obese too? There should be some mention of how BMI is categorised.

Reviewer #4: I would like to make a couple of minor suggestions. They are listed below:

The authors have incorporated the most of the previous comments into the manuscript.

But, some additional comments authors should be considered.

Introduction

In line numbers 52 and 53, the authors describe the prevalence of child marriage in the Dominican Republic and Brazil, how it is justified, and whether or not both countries have the same socioeconomic conditions as Bangladesh. It would be better to compare the Dominican Republic and Brazil to Pakistan and Indonesia, where the majority of the population adheres to the Muslim religion.

Line 89, the authors have written the putting them in the thin group, what is the thin group, the sentence is unclear.

Results

Still, Figure is not a conventional graph; please see the other published paper graphs.

In the result section, the authors have written that the COR but written Adjusted AOR, the author can write same AOR instead of adjusted AOR.

7. PLOS authors have the option to publish the peer review history of their article (what does this mean?). If published, this will include your full peer review and any attached files.

Reviewer #1: No

Reviewer #3: No

Reviewer #4: No

---

## [Author Response · Author response to Decision Letter 2]

1 Jan 2022

Dear Srinivas Goli, Ph.D.

Academic Editor

PLOS ONE

Thank you very much for providing another chance and valuable comment and feedback to improve the quality as well as suitability for publication of our manuscript. All revised texts are in “red” color. 

All necessary files are uploaded to the journal system. 

Response to Reviewer 1 comments:

1. Thank you very much for carefully checking the manuscript. Actually, the Y-axis level was wrong in Figure 1. Now, Figure 1 is corrected. Page: 9

2. Thank you very much. According to your suggestion, we revise the manuscript and add a figure (Figure 2) for presenting the prevalence of diarrhea among children of various ages by women's age at marriage. Page: 10-11

3. Thanks. We revise the texts. Page: 12

4. Thanks. We add a Supplementary Table (S1 Table) in the revised manuscript. S1 Table

Response to Reviewer 3 comments: 

Thank you very much again for carefully checking the manuscript and your feedback. It will help to improve the quality of the manuscript. 

Page: 1, 7, 9, 10

Thanks. We have checked and fixed the grammatical issues. Article “The” has been removed from some places. Page: 2-16

Thanks, we appreciate your concern. In Table 1, we add the range of BMI for making categories. Page: 8

Response to Reviewer 4 comments: 

Thank you very much for your appreciation and insightful feedback. We believe that your comments help to improve the quality of the manuscript. 

We revise the manuscript considering the information of Muslim countries like Pakistan, Indonesia, and Afghanistan. 

We add the BMI range for the “thin” group in line 89. Page: 2, 3, 

Thanks for carefully checking the manuscript. We replace Figure 1 and incorporate all your suggestions in the revised version of the manuscript. Page: 9, 11-13

In conclusion, the revised version of the manuscript has been produced as per the review outcomes. So, we hope that you will be happy to see this greatly improved version. Once again, we would like to thank you all for your dedication, professional services, and cooperation.

---

## [Editor Report · Decision Letter 3]

6 Jan 2022

PONE-D-21-24855R3Child marriage and its association with morbidity and mortality of under-5 years old children in BangladeshPLOS ONE

Dear Dr. Hossain,

Thank you for submitting your manuscript to PLOS ONE. After careful consideration, we feel that it has merit but does not fully meet PLOS ONE’s publication criteria as it currently stands. Therefore, we invite you to submit a revised version of the manuscript that addresses the points raised during the review process.

ACADEMIC EDITOR: I am sending back the paper to you for a minor revision. Please provide the full regression tables in the supplementary file and attach with the paper. Also, in predictors variables, your using both 'socio-economic and demographic' factors but why you have written only 'demographic factors' in column heading of the Tables. The logistic regression equation is looks like copy paste from the  reference books. Can you modify the equation by placing the "Y" variable name, and the key Predictor (X) name in the equation. For example,  Y(child health) = alpha + beta.x(child marriage).........xi+error term. Also, please read the paper for language errors once again. Although, I am recommending this paper because it is about child marriage, but the analyses are not very impressive to me. 

We look forward to receiving your revised manuscript.

Kind regards,

Srinivas Goli, Ph.D.

Academic Editor

PLOS ONE

Journal Requirements:

Additional Editor Comments:

I am sending back the paper to you for a minor revision. Please provide the full regression tables in the supplementary file and attach with the paper. Also, in predictors variables, your using both 'socio-economic and demographic' factors but why you have written only 'demographic factors' in column heading of the Tables. The logistic regression equation is looks like copy paste from the reference books. Can you modify the equation by placing the "Y" variable name, and the key Predictor (X) name in the equation. For example, Y(child health) = alpha + beta.x(child marriage).........xi+error term. Also, please read the paper for language errors once again. Although, I am recommending this paper because it is about child marriage, but the analyses are not very impressive to me.
---

## [Author Response · Author response to Decision Letter 3]

6 Jan 2022

Dear Srinivas Goli, Ph.D.

Academic Editor

PLOS ONE

Thank you very much for providing another chance and valuable comment and feedback to improve the quality as well as suitability for publication of our manuscript. 

We add a supplementary Table (S2 Table) to summarize the results of logistic regression. 

We replace “demographic factors” by “Socio-economic and demographic characteristics” in Tables. 

The equation of logistic regression is revised as per feedback. 

All revised texts are in “red” color. Page: 6-8, 13, S1 Table, S2 Table

All necessary files are uploaded to the journal system. 

We would like to thank you all for your dedication, professional services, and cooperation.

Best Regards,

Hossain

---

## [Editor Report · Decision Letter 4]

10 Jan 2022

Child marriage and its association with morbidity and mortality of under-5 years old children in Bangladesh

PONE-D-21-24855R4

Dear Dr. Hossain,

We’re pleased to inform you that your manuscript has been judged scientifically suitable for publication and will be formally accepted for publication once it meets all outstanding technical requirements.

Kind regards,

Srinivas Goli, Ph.D.

Academic Editor

PLOS ONE

Additional Editor Comments (optional):

Now, this paper is ready for publication. I am recommending it.
---

## [Editor Report · Acceptance letter]

31 Jan 2022

PONE-D-21-24855R4 

Child marriage and its association with morbidity and mortality of under-5 years old children in Bangladesh 

Dear Dr. Hossain:

I'm pleased to inform you that your manuscript has been deemed suitable for publication in PLOS ONE. Congratulations! Your manuscript is now with our production department. 

Kind regards, 

on behalf of

Dr. Srinivas Goli 

Academic Editor

PLOS ONE